# A universal molecular control for DNA, mRNA and protein expression

Helen M. Gunter[1,2,3], Scott E. Youlten[4,5,6], Andre L. M. Reis [7,8,9], Tim McCubbin [1,3], Bindu Swapna Madala[5,8], Ted Wong[5], Igor Stevanovski [7,8], Arcadi Cipponi[5,6], Ira W. Deveson[7,8,9], Nadia S. Santini[10], Sarah Kummerfeld[5,6], Peter I. Croucher [5,6], Esteban Marcellin [1,3] & Tim R. Mercer [1,2,3,5] ✉

The expression of genes encompasses their transcription into mRNA followed by translation into protein. In recent years, next-generation sequencing and mass spectrometry methods have profiled DNA, RNA and protein abundance in cells. However, there are currently no reference standards that are compatible across these genomic, transcriptomic and proteomic methods, and provide an integrated measure of gene expression. Here, we use synthetic biology principles to engineer a multi-omics control, termed *pREF*, that can act as a universal molecular standard for next-generation sequencing and mass spectrometry methods. The *pREF* sequence encodes 21 synthetic genes that can be in vitro transcribed into spike-in mRNA controls, and in vitro translated to generate matched protein controls. The synthetic genes provide qualitative controls that can measure sensitivity and quantitative accuracy of DNA, RNA and peptide detection. We demonstrate the use of *pREF* in metagenome DNA sequencing and RNA sequencing experiments and evaluate the quantification of proteins using mass spectrometry. Unlike previous spike-in controls, *pREF* can be independently propagated and the synthetic mRNA and protein controls can be sustainably prepared by recipient laboratories using common molecular biology techniques. Together, this provides a universal synthetic standard able to integrate genomic, transcriptomic and proteomic methods.

Next-generation sequencing (NGS) can measure DNA and mRNA abundance, whilst mass spectrometry (MS) can measure protein abundance. Over the past decade, these high-throughput technologies have provided a detailed profile of the genome, transcriptome and proteome within a cell. However, despite their importance, sequencing and proteomic technologies suffer from technical errors and biases that confound the accurate analysis of DNA, mRNA and protein abundance[1,2]. Due to these limitations, our current understanding of gene expression is often descriptive, and a truly quantitative understanding is lacking.

Reference standards comprise well-characterised materials that can be used to calibrate molecular methods. Molecular standards can

[1]Australian Institute of Bioengineering and Nanotechnology, The University of Queensland, Brisbane, Queensland, Australia. [2]BASE mRNA Facility, The University of Queensland, Brisbane, Queensland, Australia. [3]ARC Centre of Excellence in Synthetic Biology, The University of Queensland, Brisbane, Queensland, Australia. [4]Department of Genetics, Yale University School of Medicine, New Haven, CT 06510, USA. [5]Garvan Institute of Medical Research, Sydney, New South Wales, Australia. [6]St Vincent's Clinical School, University of New South Wales, Sydney, New South Wales, Australia. [7]Genomics and Inherited Disease Program, Garvan Institute of Medical Research, Sydney, New South Wales, Australia. [8]Centre for Population Genomics, Garvan Institute of Medical Research and Murdoch Children's Research Institute, Sydney, New South Wales, Australia. [9]School of Electrical and Information Engineering, University of Sydney, Sydney, New South Wales, Australia. [10]Centro Nacional de Investigación Disciplinaria en Conservación y Mejoramiento de Ecosistemas Forestales, INIFAP, Ciudad de México 04010, Mexico. ✉e-mail: t.mercer@uq.edu.au

act as qualitative controls that evaluate the detection of DNA or protein sequences, as well as quantitative controls that evaluate the abundance of genomes, mRNAs or proteins[3,4]. These controls can measure the impact of technical variation that confounds NGS and MS experimental accuracy and reliability[5,6].

The most commonly used standard in molecular biology is the bacteriophage *PhiX-174* genome. First sequenced in 1974[7], the *PhiX-174* genome is widely used as a DNA standard for molecular cloning and

sequencing, where it is used to measure error rates and improve the diversity of low complexity libraries. However, despite serving admirably as a standard for almost fifty years, the *PhiX-174* genome was selected serendipitously and suffers from several limitations, including a poor representation of the most error-prone sequences that are challenging for NGS.

More recently, a diverse range of DNA, RNA and protein controls have been developed to measure genomic, transcriptomic and

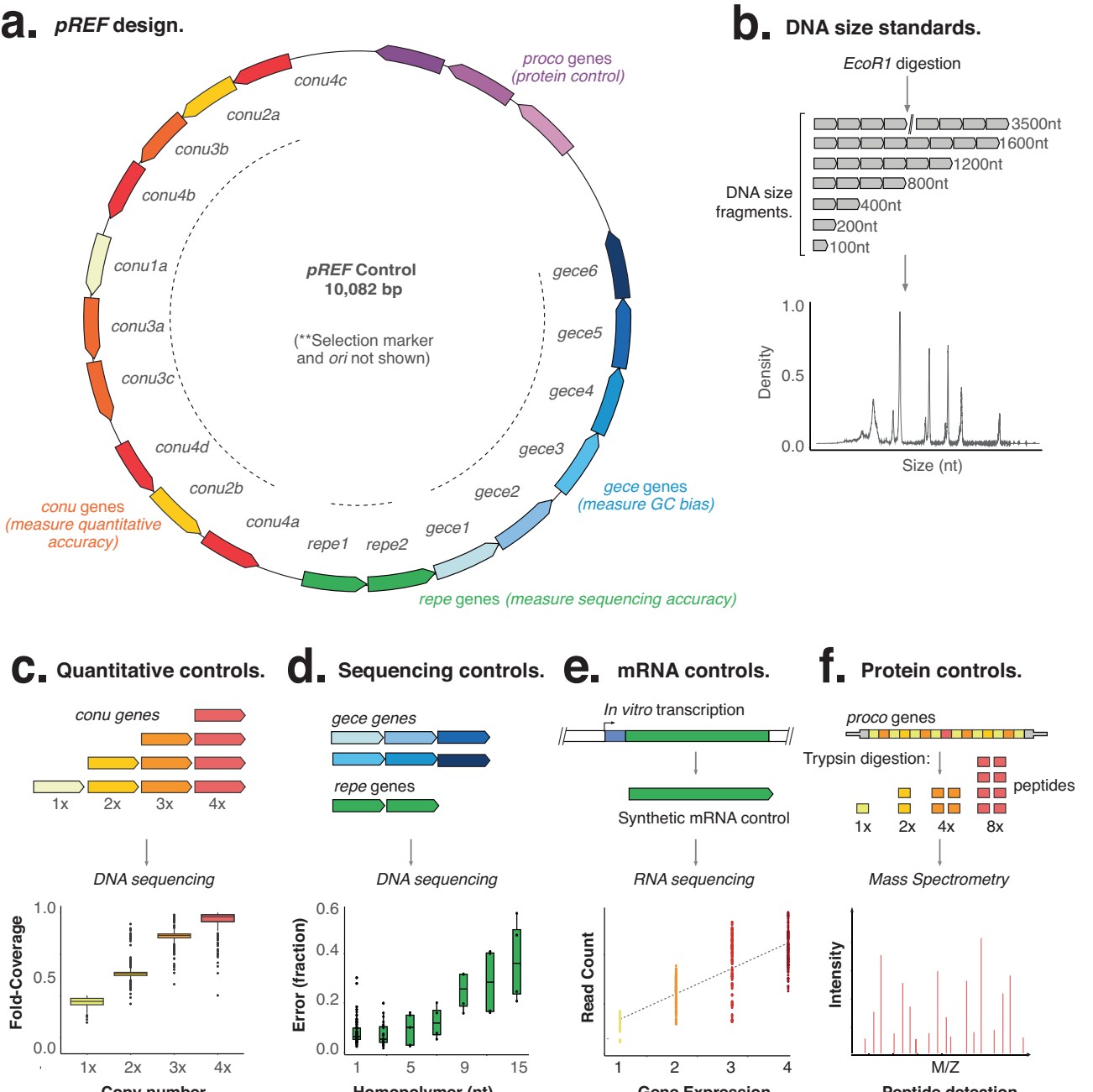

**Fig. 1 | Design of *pREF*. a** Overview of *pREF* shows the organization of synthetic *conu, gece, repe* and *proco* genes that are suitable for DNA and RNA sequencing, and mass spectrometry. **b** The digestion of *pREF* (with EcoRI) generates a DNA fragment size ladder (nt = nucleotides). **c** *conu* genes are represented at multiple copy-numbers in *pREF* so that when sequenced, they form a staggered reference ladder able to measure quantitative features of DNA and RNA sequencing libraries ($n = 1$ biologically independent samples). Box plot extends from 25th to 75th percentiles, centre line is the median, and whiskers cover the 10th and 90th percentiles. **d** *gece*

and *repe* genes can act as sequencing controls that measure accuracy at difficult GC-rich or repetitive sequences, respectively ($n = 1$ biologically independent samples). Box plot extends from 25th to 75th percentiles, centre line is the median, and whiskers cover the 10th and 90th percentiles. **e** In vitro transcription of *conu, gece, proco* and *repe* synthetic genes generate matched mRNA controls for use in RNAseq experiments. **f** In vitro translation of *proco* genes generates synthetic protein controls for use in proteomic experiments. Source data are provided in a Source Data File.

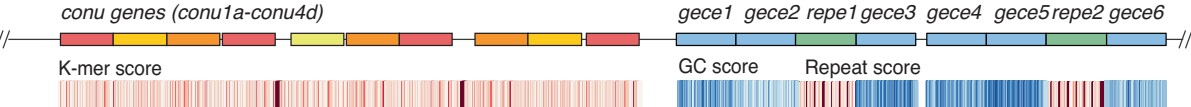

**a.** *pREF* reference annotations.

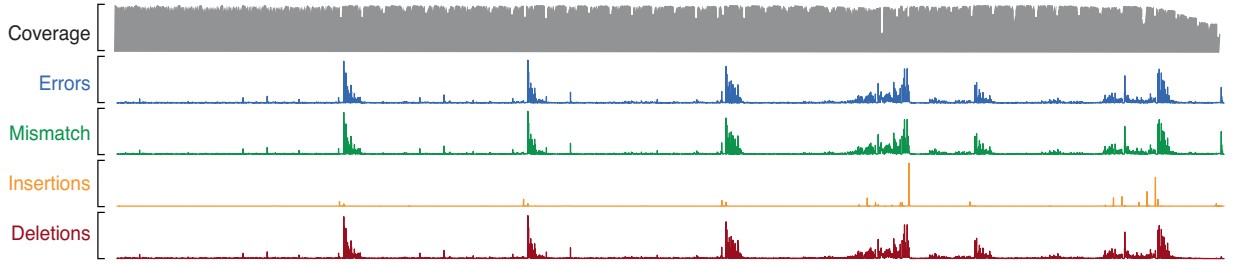

**b.** Illumina short-read sequencing.

**c.** Oxford Nanopore long-read sequencing.

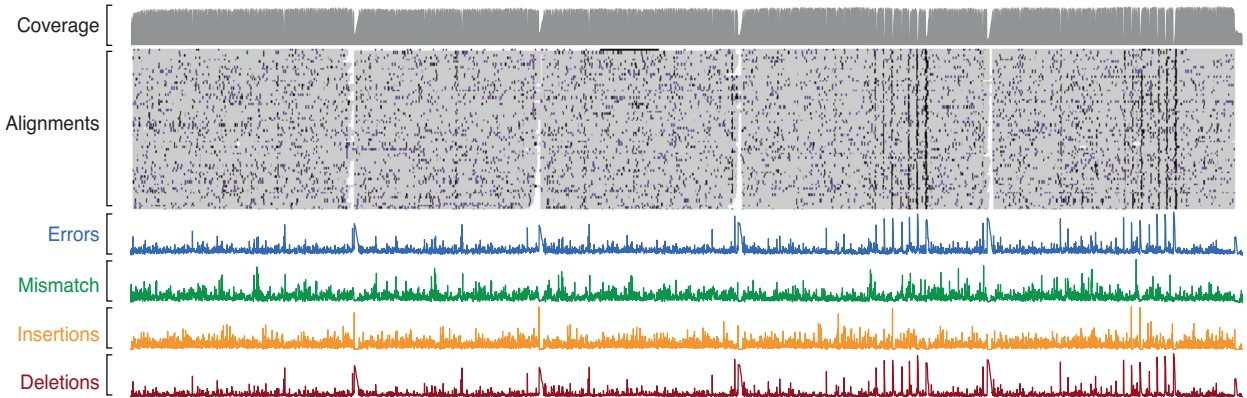

**d.** K-mer analysis of sequencing accuracy using *pREF*.

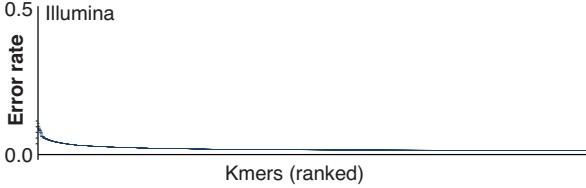
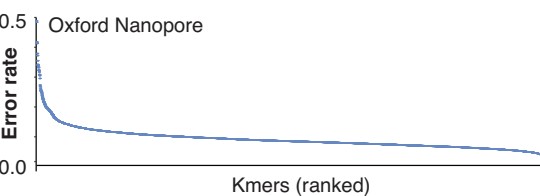

**e.** Error profiles at individual k-mers.

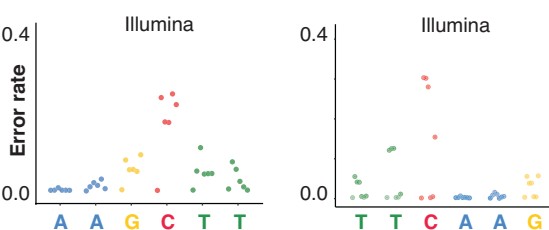
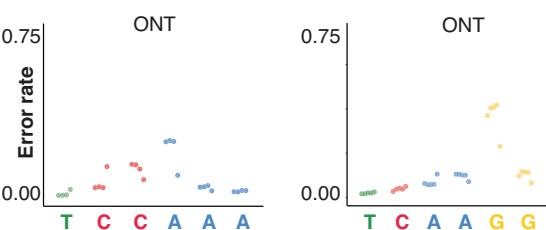

**Fig. 2 | Using *pREF* to measure errors in Illumina and ONT sequencing.**
**a** Annotations of *conu, gece* and *repe* genes in *pREF*, with heat maps showing k-mer coverage, GC-content and repetitiveness. Error profile of (**b**) Illumina and (**c**) ONT alignments across the *pREF* sequence. **d** Histogram of individual k-mers ranked according to error rate for Illumina and ONT sequencing. **e** Individual error profiles for selected k-mers (ONT = Oxford Nanopore Technologies). **a**–**e** A single replicate from each sequencing method was used to create each plot. Source data are provided in a Source Data File.

proteomic technologies[8]. Well-characterised human genomes, such as NA12878, have been used to improve the accuracy and reliability of genome sequencing. Synthetic RNAs that are 'spiked-in' to samples and analysed as internal controls have been developed to measure the sensitivity and accuracy of RNA sequencing[9]. Finally, protein standards that yield a suite of peptides with known abundance and performance and are often needed to calibrate mass spectrometric and electro-phoretic analysis conditions. Together, this diverse range of molecular standards has become increasingly important for the analysis of complex datasets generated by high-throughput NGS and MS methods.

In recent years, multi-omics studies have attempted to integrate different NGS and MS methods to provide a comprehensive profile of gene expression. However, despite the increasing use of multi-omics approaches, there are currently no universal standards that can be used to evaluate DNA, RNA and protein measurements. A universal molecular standard that could work across genomic, transcriptomic and proteomic methods could provide a unifying standard for gene expression, and integrate diverse data-sets from multi-omics projects.

Inspired by the *PhiX-174* genome and recent advances in synthetic biology, DNA assembly and de novo proteins, we rationally designed a synthetic control, termed *pREF*, to serve as a universal molecular standard. We show how *pREF* can be used to evaluate the qualitative and quantitative accuracy of DNA sequencing. Furthermore, *pREF* encodes synthetic genes that can be in vitro transcribed into matched mRNA controls, and in vitro translated into protein controls. We show how these genes can act as spike-in controls for DNA and RNA sequencing, as well as MS experiments. Unlike other reference standards, *pREF* can also be propagated and modified by recipient laboratories, and used to prepare matched mRNA and protein controls independently, thereby enabling the decentralized distribution of *pREF*. Therefore, we provide *pREF* as a first-generation synthetic control to improve the reliability, accuracy and integration of genomic, transcriptomic and proteomic technologies.

## Results

### Design of synthetic *pREF* control

We first designed a synthetic control, that we term *pREF*, that encodes 21 synthetic genes encompassing a wide range of nucleotide sequences, GC content and other difficult features (Fig. 1a). It has been deposited with Addgene, who are responsible for it's storage and distribution. *pREF* includes a full representation of all possible 6-mers (excluding unintended Restriction Enzyme recognition sequences), thereby providing a comprehensive evaluation of sequencing accuracy under different nucleotide contexts (Fig. S1a). *pREF* also includes eight genes containing difficult sequences, including six *gece* genes (gc-content), that encompass different percentages of GC content from 21% to 65%, as well as two *repe* genes (repeat) that include repeats ranging from 6 to 18 nt in length (Fig. 1d). A comparison of these synthetic genes shows they provide a greater representation of nucleotide diversity, repeat sequences, and GC content than the *PhiX-174* genome (Fig. S1a-b). Furthermore, the synthetic genes have no homology (greater than 18nt) to natural gene sequences included in the NCBI nr/nt database and can be easily distinguished from natural RNA or DNA samples, allowing the use of *pREF* as a spike-in control in the study of any organism included in the NCBI nr/nt database[10].

Synthetic control genes are flanked by a range of restriction enzyme sites enabling linearization of genes into DNA fragments of known sizes and sequence (Fig. 1b). For example, digestion with EcoRI generates a ladder of DNA fragments that range from 100 – 3500 nt, which are suitable for use as a DNA spike-in. Synthetic control genes are also preceded by T7 and Sp6 promoters that enable in vitro tran-scription and are followed by a 30nt poly-adenine tract (see Fig. 1e). These transcriptional units can be liberated by cleavage with HindIII, and transcribed using T7 RNA polymerase to yield transcripts of ~1500

nt in length. Alternatively, transcripts can be digested with BamHI and transcribed using Sp6 RNA polymerase to yield transcripts of varying lengths (see Methods).

We also designed a suite of *conu* genes (copy-number genes) to act as quantitative controls (Fig. 1c). The *conu* genes are repeated at different copy-numbers (1x, 2x, 3x and 4x) to form "paralogous gene families". Due to their repeated copy-numbers, the sequenced read count for each *conu* gene will be proportional to its copy-number, and in fixed and constant ratio to the other *conu* genes. Therefore, a comparison of observed *conu* gene count to expected copy-number generates a staggered, graduated reference scale that is able to mea-sure quantitative performance of NGS and MS methods[11].

Additionally, we designed three *proco* genes (protein control) that can provide quantitative reference standards for proteomic analysis. The *proco* genes each encode an open-reading frame with known amino-acid sequences, as well as a Shine-Dalgarno and 5' and 3' untranslated sequences to enable efficient in vitro translation (Fig. 1f). The *proco* genes can be translated to form protein spike-in controls, and trypsin digestion of proco proteins liberates peptide sequences of known size, charge and retention time for the calibration of mass spectrometric experiments.

The *pREF* sequence also encodes a pMK-RQ backbone sequence containing an antibiotic resistance gene (*Kanamycin*) and an origin of replication (Ori). These regulatory elements enable ongoing and sus-tainable production of *pREF* by independent laboratories through transformation and propagation in a stable *E. coli* line and purification using standard plasmid preparation techniques (see **Methods**). This ability to reproduce and propagate *pREF*, and independently prepare matched mRNA and protein controls using standard molecular biology techniques distinguishes *pREF* from other molecular standards[12–15].

### Measuring next-generation sequencing accuracy with *pREF*

*pREF* is designed to be used as a molecular standard in NGS, where it provides a detailed and comprehensive evaluation of NGS accuracy and performance. To demonstrate this approach, we first sequenced four technical replicates of *pREF* using Illumina short-read and Oxford Nanopore (ONT) long-read sequencing (Fig. S14).

We first measured sequencing accuracy across the *conu, repe* and *gece* genes, which include all 6-mers, providing a detailed signature error profile for the Illumina DNA libraries (Fig. 2a,b, Fig. S2a-c, Fig. S3a). We observed a mean 0.009 error-rate with a wide 200-fold range in error rate between the least accurate (0.1200 for TCTTGT) and most accurate k-mers (0.0006 for ACTAGT; Fig. S3a) in one replicate. This sequencing error profile is systematic, with little varia-tion in performance between three further technical replicate libraries (SD 0.0014; Fig. S3b). For comparison, we also sequenced the *PhiX-174* genome, which exhibited similar sequencing accuracy within the nar-rower window of represented k-mers (Fig. S3c, S1a-b). *pREF* encodes genes with extensive repeats (*repe* genes), and strong GC biases (*gece* genes) that enable evaluation of NGS accuracy and performance at these difficult sequences. We found that the presence of repeats had a complex impact on Illumina sequencing accuracy, with higher rates of deletion and mismatch errors than insertion errors, and an unex-pectedly higher error rate for shorter repeats (Fig. S2b-c, Fig. S4a). By contrast, GC-rich regions in *gece* genes were dominated by mismatch transversion errors (Fig. S2b, Fig. S4b).

ONT sequencing provides real-time, long read and single-molecule sequencing, but suffers from errors that can confound analysis[3]. We evaluated the error profile of ONT sequencing using *pREF* (Fig. 2c, Fig. S5a-c), which showed a higher mean error frequency than Illumina sequencing (0.0613; SD 0.03455) (Fig. S6a), with a 33-fold difference between the most and least accurate k-mers (from 0.0167 to 0.55794) (Fig. 2d). Comparing ONT and Illumina sequencing showed different per nucleotide error rates within different k-mers (Fig. 2e). We also evaluated the ONT sequencing error profile at repeats within

## a. Schematic illustration of quantitative *conu* genes.

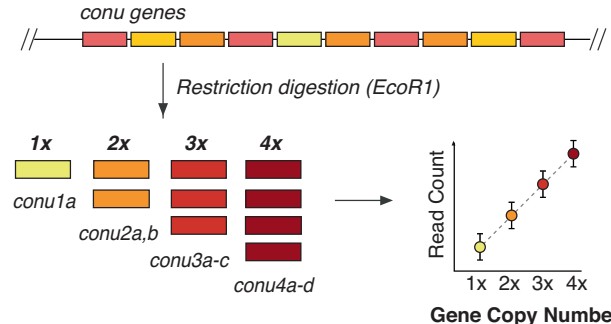

## b. Quantitative distribution of *conu* genes.

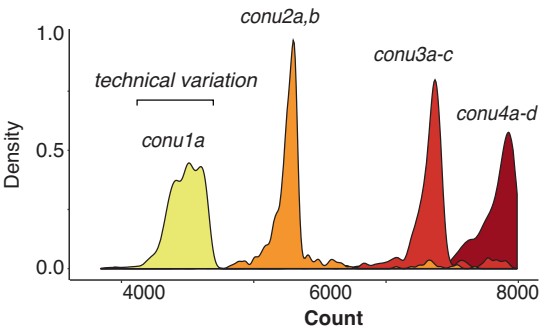

## c. Measuring quantitative accuracy with *pREF* in DNA and RNA sequencing.

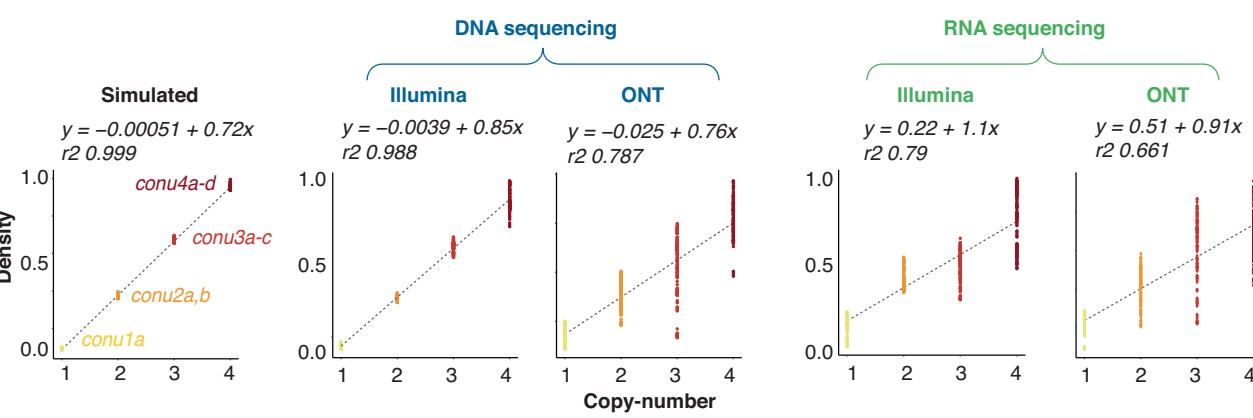

## d. Measuring technical variation with *pREF* in DNA and RNA sequencing.

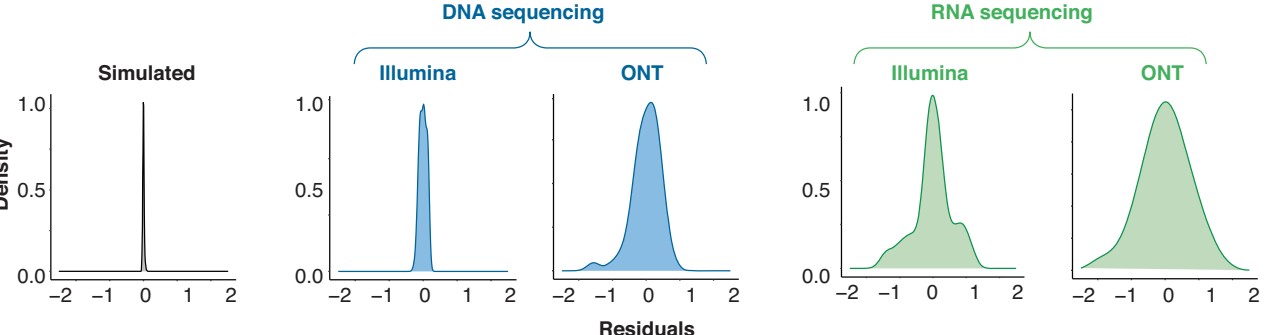

**Fig. 3 | Measuring the quantitative accuracy of synthetic *conu* genes.**
**a** Schematic diagram illustrates the design and use of *conu* genes as quantitative controls. **b** Density histogram from k-mer counts for *conu* gene families illustrates the distribution of technical variation in 31-mer normalised read count, calculated using a sliding window approach. Bounds of technical variation are a visual representation of read count variation and are not based on a statistical calculation.

**c** Quantitative accuracy of simulated, DNA and RNA sequencing using Illumina and ONT sequencing technologies as measured from *conu* genes (ONT = Oxford Nanopore Technologies). **d** Density plots show the spread of technical variation in *conu* read counts in DNA and RNA libraries, prepared for ONT and Illumina sequencing. Source data are provided in a Source Data File.

the *repe* genes (Fig. S6b,c), finding that error rate has a positive linear relationship with repeat length (Pearson's correlation ($R^2$) = 0.93) up to 0.3949 for an 18-nt homopolymer. The majority (93%) of errors at repeats were deletions, resulting in reads being ~16% shorter than expected (Fig. S6c).

Sequencing of *gece* genes also showed that ONT sequencing errors scaled linearly with GC content ($R^2$ = 0.9442), from 0.0454 at 21% GC, up to 0.0728 at 65% GC content (Fig. S6d). Unlike Illumina sequencing, we observed similar proportions of all error types in our

*gece* genes (compare Fig. S2c **to** Fig. S6d). This demonstrates how direct analysis of the *pREF* sequence provides a comprehensive evaluation of different reagents, instruments and libraries used in NGS.

**Measuring quantitative accuracy using *pREF conu* genes**
*pREF* includes repeated *conu* genes which, when sequenced, generate a staggered ladder that can evaluate the quantitative accuracy and variation in NGS libraries (Fig. 3a). To demonstrate this approach, we analysed the read counts across 31-mer sliding windows for all *conu*

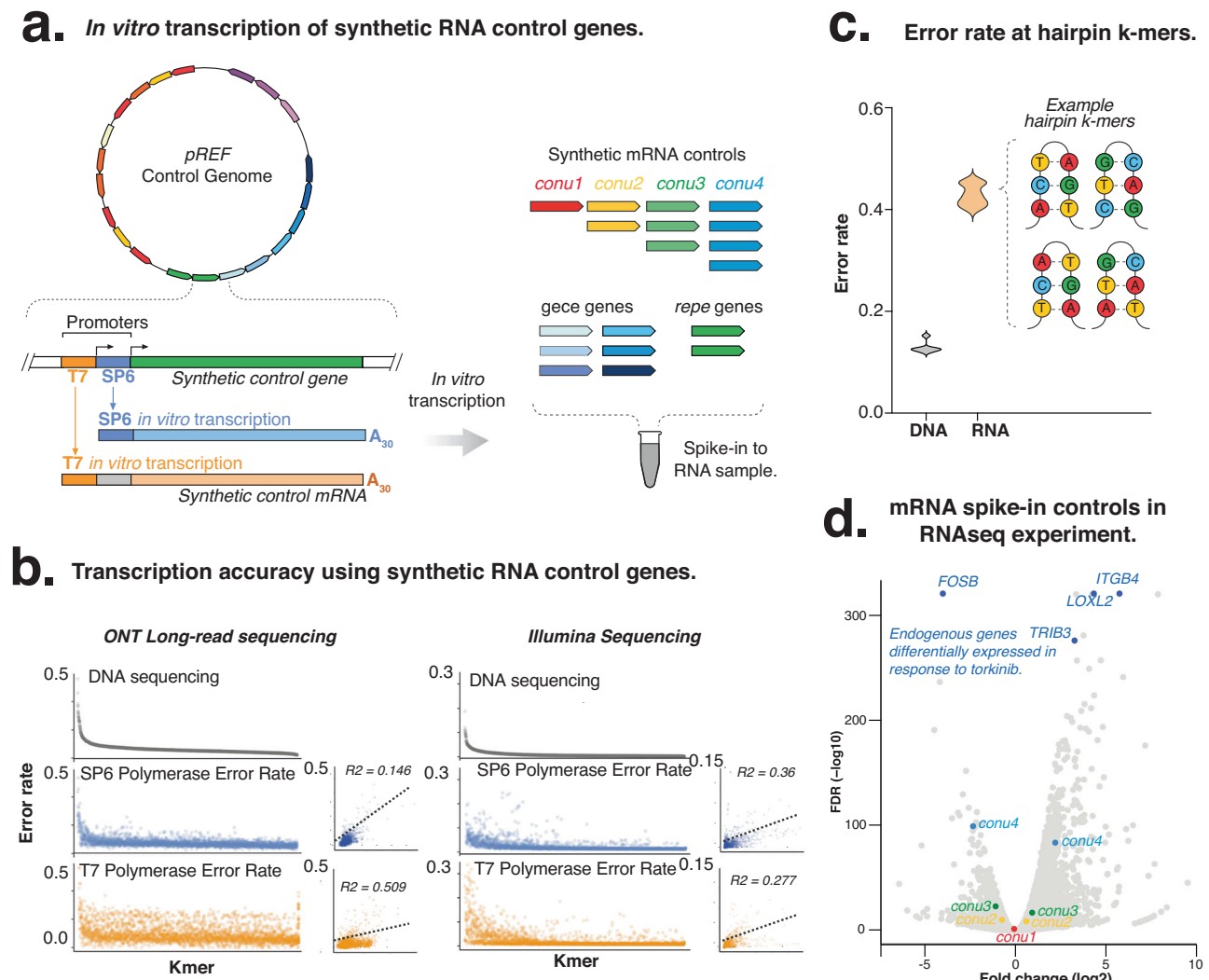

**Fig. 4 | In vitro transcription of *pREF* mRNA controls. a** Synthetic control genes are preceded by a T7 promoter that enables in vitro transcription into matched mRNA controls. **b** Scatter plots indicate the fold-differences in k-mer sequencing error rates between RNA and DNA libraries for Illumina and ONT sequencing. **c** Violin plot illustrates the enrichment of sequencing errors at k-mers that form hairpins. **d** Use of synthetic *conu* gene during differential gene analysis of lung adenocarcinoma cells with torkinib. The synthetic RNA controls (coloured points) indicate the accuracy for detecting fold change differences in gene expression (grey) between treated and untreated cells. Source data are provided in a Source Data File.

genes in Illumina and ONT sequenced DNA libraries (Fig. S14, Fig. 3b,c, Fig. S7a). For the Illumina DNA libraries, we observed a strong quantitative correlation between *conu* gene copy number and read count (Pearsons $R^2 = 0.9875$, Fig. 3c). Consistent ratios were observed between the successive *conu* genes (mean = 1.991, SD = 0.1490). However, this relationship was weaker for ONT DNA sequencing, due to the lower quantitative accuracy of these libraries ($R^2 = 0.79$, SD = 0.1490, Fig. 3c, Fig. S7a).

We next used *pREF* to investigate the impact of technical variation introduced during the experimental steps of library preparation and sequencing. This technical variation can be measured by the symmetric, unimodal distribution of k-mer counts for *conu* genes (Fig. 3d). In a simulated library, which excludes experimental variation, we measured only minor technical variation due to random sampling error (coefficient of variation (CoV) = 1.26%, SD = 0.0807). By contrast, Illumina libraries showed additional variation due to experimental steps (CoV = 5.63%, SD = 0.1922), whilst the ONT libraries showed the highest degree of variation during their preparation (CoV = 12.56%, SD = 0.6537). This provides a useful estimate of variation within an NGS library, even in the absence of experimental replicate libraries.

Given this ability to estimate technical variation, *pREF* can be used to normalize technical differences between different libraries, whilst retaining biological differences[16]. To demonstrate this, we spiked *pREF* into two mock microbial communities[13]. The two mock communities, which contain synthetic microbial sequences at known concentrations, were then sequenced in triplicate using ONT DNA sequencing (Fig. S14). The observed differences in abundance were plotted against their expected abundances, demonstrating the technical variation introduced by sequencing (Fig. S7b) Sequencing data were normalized using removal of unwanted variation (RUVg), with the *pREF* genes providing negative scaling factors[16]. We found that RUVg normalization using *pREF* as a spike-in control improved normalization compared to either unnormalized or Upper Quartile (UQ) normalisation methods[17] (see Fig. S7c). Accordingly, the use of *pREF* for normalization resulted in improved detection of known fold-change differences in microbial abundance between communities, and demonstrates how *pREF* can be routinely used as an internal control to normalise technical variation and improves detection of fold-change differences.

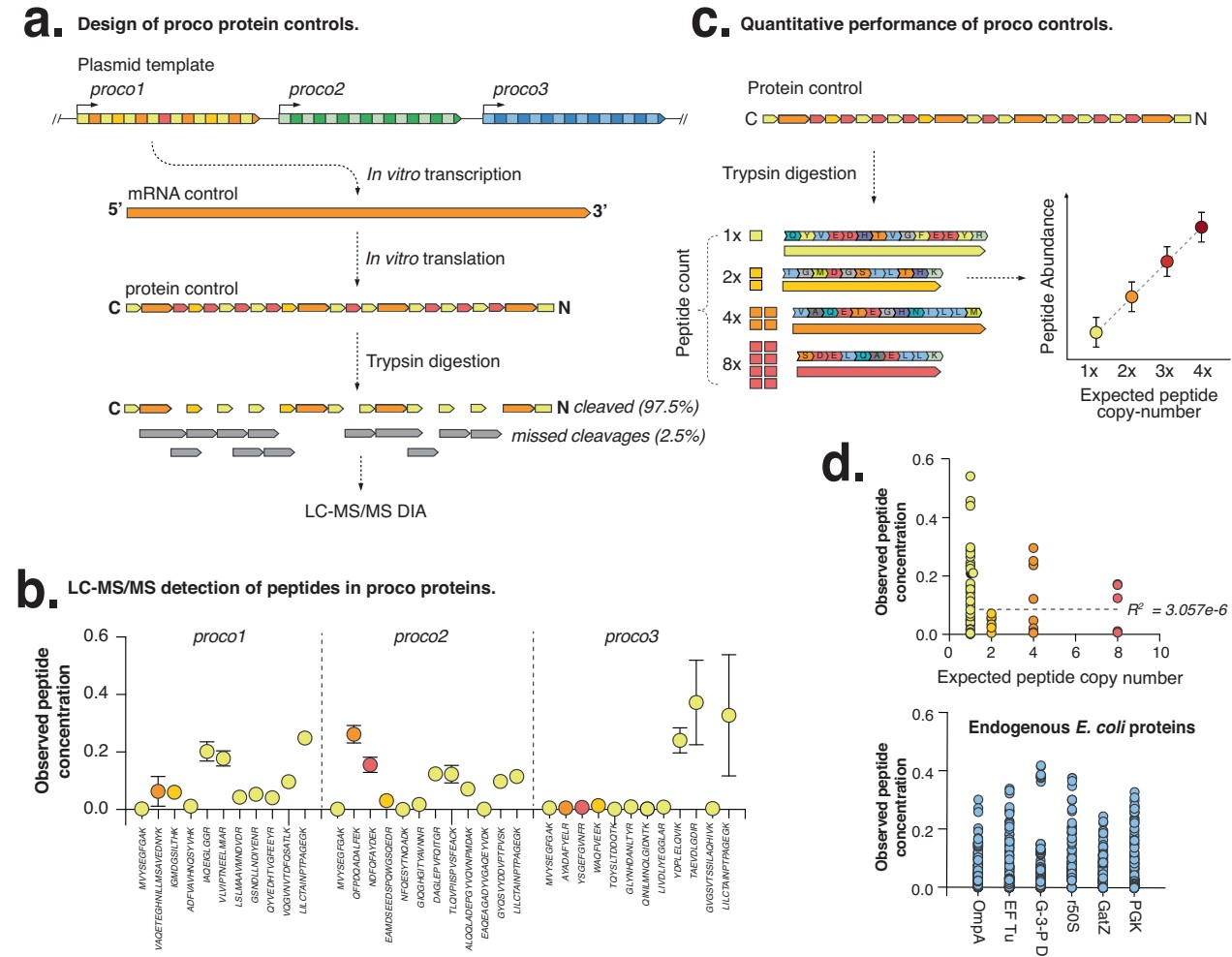

**Fig. 5 | In vitro translation of synthetic proco protein controls. a** Schematic illustrates the design of *proco* genes that are translated to form protein controls. Trypsin digestion of the proco proteins then liberates control peptides of differing size, charge and retention time, enabling the calibration of LC-MS/MS. **b** Quantification of each fully cleaved proco peptide. Relative peptide abundance is measured by the proportion of detected peptides relative to all peptides in the proco protein. Data are presented as mean values +/- SD (n = 3 biologically independent samples). **c** Schematic diagram indicates how peptides are also present at differing copy-number, thereby forming a staggered quantitative reference ladder for evaluating quantitative performance of proteomic experiments. Data plots are for illustrative purposes only, and are not based on Mass Spectrometric measurements. **d** Measurement of relative peptide abundance for proco proteins and housekeeping *E. coli* proteins (where each peptide is expected to be in equal abundance) in replicate (n = 3). Source data are provided in a Source Data File.

## Use of synthetic mRNA controls during RNA sequencing

The synthetic genes encoded within the *pREF* sequence can be in vitro transcribed to generate a suite of matching mRNA controls for use in RNA sequencing experiments (Fig. 4a). To demonstrate this, we performed in vitro transcription using T7 RNA polymerase to generate a matched suite of mRNA controls. We then prepared mRNA controls as cDNA libraries for both Illumina and ONT sequencing (see Methods).

We first evaluated the accuracy of RNA sequencing for Illumina and ONT sequencing across *repe*, *conu* and *gece* genes (Fig. S14). We observed markedly higher error rates and greater technical variation in RNA compared to DNA (Fig. 4b, Fig. S8a-c). For Illumina RNAseq libraries, we observed a mean k-mer error rate of 0.0154, which is 2.6-fold higher than the corresponding DNA sequencing (Fig. S9a). We also measured the quantitative accuracy of *conu* gene expression, showing that correlation in *conu* gene expression was lower for RNA ($R^2 = 0.79$), than for DNA ($R^2 = 0.9875$, Fig. 3c). Similar to our Illumina sequencing results, we observed higher error rates for ONT RNA compared to DNA (Fig. S8a-c, S9a-b). Moreover, we observed a higher error rate and technical variability for ONT RNAseq than for Illumina RNAseq (Figs. 3d, 4b).

Analysis of mRNA controls also showed the impact of RNA secondary structure on sequencing accuracy. Examination of RNAseq error rates showed that short, inverted repeats were particularly error-prone (such as an error rate of 0.555 for ACTAGT, or 0.4851 for CTGCAG kmers) for RNA sequencing. This is likely to be due to the formation of hairpin loops that may impact transcription by RNA polymerase, as RNA inverted repeats exhibited an 83-fold higher error rate than corresponding DNA sequencing (Fig. 4c). We also observed enrichment for errors in *repe* and *gece* genes in ONT RNA sequencing datasets, relative to complementary DNA sequencing results (Compare Fig. S6c-d to S10a-b). This indicates the additional exacerbation of errors at repetitive and GC-biased sequences during RNAseq library preparation and sequencing.

We next used the *pREF* control genes as reference standards to measure the enzyme performance of either Sp6 or T7 RNA polymerases (Fig. S14). We first performed in vitro transcription using either Sp6 or T7 RNA, and then ONT sequenced in triplicate. Data were analysed using per-nucleotide normalisation against the matched DNA sequencing libraries to normalise for sequencing-specific errors (**see Methods**). We found that Sp6 polymerase (median error rate =

0.004579, SD 0.03401) performed markedly better than T7 polymerase, which was less accurate and exhibited a higher variance (see Fig. 4b). Closer comparison of T7 and Sp6 RNA polymerase accuracy provided a high-resolution analysis of polymerase processivity and performance. For example, we found RNA polymerase performance varies depending on the nucleotide context, with Sp6 performing more poorly on pyrimidine-rich (C,T) k-mers (1.4-fold enrichment), while performing comparatively better on purine-rich (A,G) rich k-mers (see Fig. 4b). Both Sp6 and T7 performance was impacted by the presence of repeats, with a 4.12 and 5.75 fold-enrichment in sequencing error compared to the background, respectively.

To demonstrate the use of *pREF conu* genes in an RNA sequencing experiment, we performed a drug-treatment experiment that evaluated the impact of torkinib treatment, a selective and ATP-competitive mTOR inhibitor, on gene expression in lung adenocarcinoma A549 cells (Fig. 4d, Fig. S14; see Methods). We spiked the *conu* mRNA controls into duplicate RNA samples harvested from treated and untreated cells prior to library preparation and sequencing. We first assessed the quantitative accuracy of our RNA sequencing experiment by comparing the expression *conu* spike-in mRNA controls between libraries ($R^2 = 0.8233$). As previously indicated, the *conu* mRNA controls enabled us to estimate the technical variation between libraries (CoV=18.23%) and could be used as negative scaling factors for accurate RUVg normalisation between the samples (see Methods).

RUVg can remove technical variation through factor analysis using reference standards[16]. We tested whether RUVg normalisation with *pREF* could also be used for normalisation of an RNA-seq experiment. Following normalisation with RUVg, we used the *conu* genes to benchmark the detection of fold-differences in gene expression between control and treated libraries. The endogenous genes *FOSB, LOXL2, TRIB* and *ITGB4*, were differentially expressed in response to torkinib, in a pattern that is expected from mTOR inhibition (Fig. 4d). This spike-in use of *conu* mRNA controls allows normalisation between RNA sequencing libraries, without relying on housekeeping genes, which can be highly variable[18]. This method will be particularly useful for experiments that induce global gene expression changes or with low biological replication, where negative controls and analyses of variation are not an option.

### Use of synthetic *proco* genes to improve accuracy of protein quantification

We designed synthetic *proco* genes that can be in vitro translated for use as a matched peptide controls during proteomic analysis (see Fig. 5a). The trypsin digestion of proco proteins liberates smaller peptides of differing size, charge and retention time, enabling the calibration of MS experiments. Furthermore, a subset of peptides is repeated at variable copy-numbers (2x, 4x and 8x) so that when they are digested with trypsin, they form a staggered, quantitative peptide ladder (Fig. 5a).

We first expressed the proco proteins for use in the proteomic analysis of the *E.coli* proteome (Fig. S14, see Methods). Samples were trypsin digested and analysed using LC-MS/MS, and we identified the majority of peptides encoded within the three *proco* genes, alongside 2,306 endogenously expressed *E. coli* proteins (Fig. S13). We observed high levels of digestion (97.5%, 94.5% and 99.8%) of the proco proteins, with few undigested peptides, which compared similarly to a subset of highly abundant *E. coli* housekeeping genes (a mean ~91.8% trypsin digestion) that were analysed (Fig. S13 and S11a).

We next measured peptide counts using LC-MS/MS in untargeted Data Independent Acquisition (DIA) mode, which is thought to yield accurate quantification, even in the absence of standards[19]. We first used the *proco* genes to evaluate the quantitative accuracy of LC-MS/MS DIA datasets by comparing the measured abundance of individual proco peptides relative to their expected copy numbers (Fig. 5c). We found that observed proco peptide abundance correlated poorly with

expected copy numbers ($R^2 = 3.057 e^{-06}$; Fig. 5d). These observed discrepancies in detected MS signals may be attributed to the sequence-dependent variation in ionization efficiencies between different peptides. However, despite these limitations, we observed high reproducibility in peptide quantification across three technical replicates indicating that while LC-MS/MS may not provide accurate quantification between peptides with different sequences, it does offer a reproducible and quantitative means of assessing the relative abundance of a given peptide between different samples (Fig. 5b).

To investigate the source of this variation, we evaluated whether the physicochemical properties of proco peptides confounded their quantification. We performed a multiple linear regression to compare peptide quantification against a range of predicted peptide physicochemical properties. Our analysis indicated that Molecular Weight, Hydrophobicity, Aliphatic and Instability Indexes did not confound peptide quantification. However, the Isoelectric Point and Extinction Coefficient of peptides both had a significant, albeit weak, negative correlation with peptide quantification ($p = 0.047$, $p = 0.0076$, respectively; Fig S12). Further variables such as post-translational modifications, degradation or cellular export of proteins may also have impacted peptide detection; however, their measurement is outside the scope of this experiment.

We also considered whether the observed discrepancies in LC-MS/MS quantification measured with the proco peptide controls similarly impacted the measurements of the accompanying *E. coli* proteins. To evaluate the variation in quantification, we assumed that each unique peptide within the selected *E. coli* housekeeping proteins was present once and was therefore equally abundant (Fig. S11a). Therefore, by comparing the variation in peptide quantification within *E.coli* proteins, we could estimate the quantitative accuracy of our LC-MS/MS experiment and compare it to proco controls. Similar to our proco peptide results, we observed a wide variation in the quantification of different *E. coli* peptides, with strong reproducibility for the same peptide between biological replicates (Fig. S11b-c). This demonstrates how proco proteins can evaluate and validate the quantification of protein abundance using MS methods.

## Discussion

The *PhiX-174* bacteriophage was the first genome to be sequenced and, as a result, was serendipitously selected and used as the most common molecular biology standard for almost 50 years[7]. However, given advances in DNA synthesis and synthetic biology, we designed a synthetic standard, *pREF*, that encoded control genes that could act as a superior universal standard across genomic, transcriptomic and proteomic experiments. Like the *PhiX-174* genome, we showed how *pREF* can be used as a positive control during metagenome experiments where it can evaluate sequencing accuracy and performance. However, *pREF* can also be in vitro transcribed to produce a suite of mRNA controls that measure RNA polymerase accuracy and be 'spiked-in' to RNA samples to help identify fold-change differences between genes in response to drug treatment. The *pREF* control genes can also be translated into protein controls. Digestion of these control proteins yields a suite of peptides with known characteristics that can calibrate MS experiments. Collectively, this shows how *pREF* can evaluate qualitative (sequencing error, peptide identification) and quantitative (accuracy and sensitivity) performance of genomic, transcriptomic and proteomic experiments.

Multi-omics approaches are being increasingly used to integrate NGS and MS approaches to provide a comprehensive profile of gene expression, and identify biomarkers in human diseases. However, while mRNA can be measured with NGS, and protein abundance can be measured with LC-MS/MS, these two approaches have distinct foundations and are difficult to compare. The synthetic control genes described herein represent the first set of reference standards that are both transcribed and translated, and thereby provide matched

references for both mRNA and protein measurements. Unlike existing controls such as iTRAQ labelled or SIS peptides, proco peptides permit the rapid normalisation of protein extracted from any organism (as they are not designed based on organismal homology), without the use of radiolabelled peptides[20,21]. Moreover, as the proco peptides were selected based on their previous detection by LC-MS/MS, they are not impacted by insolubility[22]. Despite these advantages, our analysis found LC-MS/MS exhibited discrepancies in quantification between individual proco proteins, and supports the well-established finding that peptides do not ionize equally in LC-MS/MS. This renders direct quantitative comparison of peptides with differing sequences infeasible and will likely require further advances, such as the implementation of machine-learning models that predict and normalize intensity response based on sequence, to improve quantitative comparisons between peptides using LC-MS/MS[23,24]. These sequence-dependent discrepancies in peptide quantification likely contribute to the poor correlation often reported between the transcriptome and proteome[1], and demonstrate how *pREF* can help integrate DNA, RNA sequencing and mass spectrometric experiments that are becoming increasingly used in multi-omic studies[25,26].

Despite these limitations, we do observe that the quantification of individual *proco* peptides is highly reproducible, providing a reliable means of assessing the relative abundance of a given protein between different samples. Within this study, we show how *pREF* can be used to provide internal scaling factors by which to perform normalization between samples. The use of universal standards is needed to ensure accurate and reliable comparisons across the increasing diversity of genomic, transcriptomic and proteomic technologies. Synthetic genes can enable normalization and interoperability between different experiments, and provide insight into quantitative cell biology. For example, we envisage clinical microbiome analyses could utilise *pREF* for quantitative scaling of microbial count data, and host transcriptome analyses, the analysis of errors that accumulate during the reverse transcription of RNA templates (used in identifying RNA viruses), and in proteomics analyses. Accordingly, we provide *pREF* as a universal molecular standard to encourage data-sharing and interoperability between high-throughput NGS and MS methods.

While *pREF* comprises, to the best of our knowledge, the first example of a synthetic control genome, we anticipate that additional second-generation synthetic control genomes will be developed for compatibility with emerging technologies. *pREF* has been deposited within open-source repositories and designed so it can be easily modified for further use with other molecular biology techniques. For example, further synthetic control genes that incorporate additional features, such as affinity tags, fluorescent peptides and CRISPR sites, can be added.

Unlike other reference standards, *pREF* has been deposited with Addgene, who will manage its' storage and distribution. Recipient laboratories can independently propagate *pREF*, and sustainably prepare synthetic mRNA or protein controls using common molecular biology protocols. As cell-free translation rates can vary due to systematic and stochastic variables[27], recipient laboratories should verify proco protein quantity prior to their use as spike-ins. Nonetheless, this decentralized use encourages wider distribution and adoption of *pREF* that can enable standardization across genomic, transcriptomic and proteomic experiments.

## Methods

### *pREF* design
We designed *pREF* to be a modular standard for multi-omics studies. It contains different sets of genes that enable the comprehensive evaluation of sequencing accuracy and technical biases in NGS and MS. First, we designed the nucleotide sequences of the genes in each of those modules. The sequences of the *conu* genes, which measure quantitative accuracy were designed with ShortCAKE, a software

package that generates the shortest sequence to cover k-mers of a specified size (we selected 6-mers). We split the ShortCAKE sequence into 4 units, corresponding to each of the *conu* genes, which was then included at known copy-numbers (1x, 2x, 3x and 4x)[28].

The sequences of the *repe* genes, which measure sequencing accuracy, were generated randomly but were designed to contain all possible homopolymers (A, C, G and T) with 6, 9, 12 and 18 nt of length. Despite the regions of low complexity, we ensured that the overall GC content of the *repe* genes was stable at approximately 44%. In contrast, the sequence of the six *gece* genes were also generated randomly, sampling a wide range of GC contents (21, 35, 45, 47, 56 and 65%).

Synthetic control genes were flanked by restriction enzyme sites, enabling the linearization and excision of genes from *pREF*, into DNA fragments of known size and sequence (Fig. S1e). Synthetic control genes were preceded by a T7 and/or Sp6 promoter that enables in vitro transcription and is followed by a 30nt poly-adenine tract and transcriptional terminators. Digesting *pREF* with EcoRI generates a staggered DNA ladder with fragments of varying sizes (100, 200, 400, 800, 1200, 1600 and 3500 nt). Digesting with HindIII, followed by in vitro transcription using T7 promoter, will generate 5 transcripts of approximately 1500 nt in length (1300 – 1747 nt). Digesting with BamHI, followed by in vitro transcription using the SP6 promoter will generate 7 transcripts of varying lengths (89, 189, 589, 848, 1233, 1710 and 2836 nt). Finally, PstI can be used to linearise *pREF* and SpeI can be used as a cloning site to add new genes.

The *proco* genes were designed from a list of Trypsin cleaved peptide sequences with diverse physicochemical properties, that can be reliably detected using MS. Twelve peptides were selected for each of the three *proco* genes. The same peptide was selected for the C terminal peptide of each gene, as it starts with a Methionine. The subsequent (unique) peptides were tiled in different copy numbers, such that the same peptide was never serially repeated. Nine peptides were included at 1x, and one peptide was included at each of 2x, 4x and 8x. The protein sequences were flanked by His-Tags, Shine-Dalgarno sites and transcription terminators, enabling their translation in *E. coli*.

### Synthesis and preparation of *pREF*
*pREF* was synthesized by a commercial vendor (ThermoFisher-GeneArt) and revived in the laboratory. *pREF* was transformed into NEB stable *E. coli* competent cells (C3040H), then grown in a 50 ml culture and later purified. The purified *pREF* pDNA was quantified using the BR dsDNA Qubit Assay on a Qubit 2.0 Fluorometer (Life Technologies) and verified on the Agilent 2100 Bioanalyzer with the Agilent High Sensitivity DNA Kit (Agilent Technologies). The *pREF* stock was then prepared as single-use aliquots and stored at −80 °C.

*pREF* has been deposited within the Addgene catalog for access and distribution. For laboratories propagating *pREF*, we recommend DNA sequencing prior to translation and its use as a sequencing standard. Although spontaneous deletion errors are observed less frequently in stable *E. coli* lines, they can occur in repetitive sequences, such as those included in *pREF*. Purified *pREF* plasmid was translated through in vitro translation with T7 and Sp6 according to manufacturer's instructions. The plasmid backbone used for *pREF* is only suitable for propagation in stable *E. coli*, and not mammalian cell lines.

### Illumina DNA sequencing *pREF*
We first sequenced neat preparations of *pREF*. Four replicate libraries were prepared using the KAPA HyperPlus PCR-based kit (Illumina) according to the manufacturer's instructions. Prepared libraries were quantified on a Qubit (Invitrogen) and verified on the Agilent 2100 Bioanalyzer with the Agilent High Sensitivity DNA Kit (Agilent Technologies). The libraries were then sequenced on a NovaSeq (Illumina). The sequencing was performed at the Kinghorn Centre for Clinical Genomics, Darlinghurst, New South Wales.

## ONT DNA sequencing *pREF*

*pREF* was linearised using restriction enzymes, and four replicate libraries were prepared for nanopore sequencing, with the LSK108 kit (1D ligation) according to the manufacturer's instructions. The resulting libraries were sequenced on a PromethION instrument, at the Kinghorn Centre for Clinical Genomics, Darlinghurst, New South Wales. Base-calling was achieved using ONT Albacore Sequencing Pipeline Software (version 1.2.6).

## In vitro transcription *pREF*

*pREF* was linearised using two different restriction enzymes, HindIII and BamHI (10 µg for each) and was in vitro transcribed using both Sp6 and T7 polymerase, respectively (ThermoFisher). The in vitro transcription reaction was performed according to manufacturer's instructions. For in vitro transcription of *pREF* in recipient laboratories, we recommend an initial DNA sequence confirmation prior to the in vitro transcription reaction.

## Illumina RNA sequencing *pREF*

For each RNA sample transcribed using Sp6- and T7 polymerase, one replicate Illumina Tru-Seq RNA-seq library was prepared according to manufacturer's instructions. Prepared libraries were quantified on a Qubit (Invitrogen) and verified on the Agilent 2100 Bioanalyzer with the Agilent High Sensitivity DNA Kit (Agilent Technologies). The libraries were then sequenced on a NovaSeq (Illumina). The sequencing was performed at the Kinghorn Centre for Clinical Genomics, Darlinghurst, New South Wales.

## ONT RNA sequencing *pREF*

Three replicate ONT libraries were prepared according to manufacturer's instructions, from cDNAs Reverse Transcribed from single Sp6- and T7 RNA samples (LSK108 kit; Oxford Nanopore). The resulting libraries were sequenced on a PromethION instrument, at the Kinghorn Centre for Clinical Genomics, Darlinghurst, New South Wales. Base-calling was achieved using ONT Albacore Sequencing Pipeline Software (version 1.2.6).

## *pREF* alignment and kmer analysis

The four replicate Illumina short-read DNA libraries were aligned to reference sequences containing the *pREF* plasmid using BWA-MEM2[29], while the two Illumina RNA libraries (transcribed with Sp6 and T7 polymerase) were aligned using bowtie2 (v2.4.0)[30]. Long-read DNA and RNA libraries generated by Oxford Nanopore sequencing were aligned to the *pREF* reference sequence using MiniMap2 (v2.17-r941)[31] with the parameters 'minimap2 -ax map-ont' optimized for Oxford Nanopore libraries. Alignment files were sorted and indexed using samtools (v1.9)[32] and pysamstats[33] were used to retrieve the coverage and specific error types, such as mismatches or insertions and deletions, for every reference sequence position. At each position, we calculated the relative frequency of mismatches, insertions and deletions, by dividing the number of reads containing each of these errors by the total read count at that position. Error rates and GC-content for 6-mer sequences across the *pREF* reference sequence were calculated based on a sliding 6-mer window with the sequencing error profile averaged across bases in each window in R (v4.0.2) using the extractList function of the IRanges R-package (v2.22.2)[34].

## *Conu* quantitative analysis

The accuracy of *conu* genes was calculated using Jellyfish 'count' (v2.2.10)[35] to quantify unique k-mers sequenced for each control gene. A k-mer length of 31 was used and counts for equivalent sequences in forward or reverse orientation were combined using the canonical option (-C). K-mer counts were normalised for each sample by dividing counts by the mean library depth across the *pREF* reference sequence. Linear models of normalised vs expected k-mer abundance were

generated using the lm function of the stats R-package (v4.0.2)[36] and compared to a simulated *pREF* library contain 1 million paired-end short reads (150 bp) generated from the *pREF* reference sequence using wgsim (v1.9)[37].

## Metagenome normalisation

Triplicate ONT libraries were prepared for samples of two mock microbial communities (A and B), which contain synthetic microbial communities at known concentrations, as described above. *pREF* was spiked into our samples prior to library preparation. Reads were aligned to a reference sequence containing Metasequins (from www. sequinstandards.com/resources) and *pREF* using Minimap2 as described above. Reads aligned to Metasequin and *pREF* control genes were quantified using HTSeq (v2.0.1)[38]. Counts corresponding to the *pREF* reference sequence were scaled to represent 1% of total library size in each sample to omit potential errors carried forward from inaccurate quantification of DNA or pipetting biases. The observed fold difference between metasequins in Mixture A and B was compared to the expected fold-change difference for these features. Differences in abundance were compared between samples without normalisation, with TMM normalisation (without *pREF*) using the R-package edgeR (v3.30.3)[39], and with RUVg normalisation using *pREF* control genes as negative scaling factors (v1.22.0)[16].

## RNAseq analysis of torkinib treatment

RNA was extracted using standard methods from lung adenocarcinoma A549 cells (purchased from ATCC; CCL-185) that were treated with torkinib, a selective and ATP-competitive mTOR inhibitor. We spiked the *conu* mRNA controls into harvested RNA from treated and untreated cells prior to duplicate TruSeq library preparation and short-read sequencing (Illumina) as described above. For recipient laboratories, we recommend sequence confirmation of the in vitro transcribed RNA prior to its use as a spike-in. The abundance of transcripts mapping to the human transcriptome (GENCODE primary assembly annotation v36, https://ftp.ebi.ac.uk/pub/databases/gencode/ Gencode_human/release_36/gencode.v36.primary_assembly. annotation.gtf.gz) or the *pREF* control genes was quantified using Kallisto (v0.46.2)[40]. Note, to construct the Kallisto index used for quantification, a modified *pREF* reference sequence was used in which each *conu* gene sequence was only represented once and additional copies of *conu* genes were masked. Counts from *pREF* transcripts were scaled to represent 1% of total library size in each sample to omit potential errors carried forward from inaccurate quantification of DNA or pipetting biases. The duplicate libraries of the control and torkinib-treated conditions were normalised with RUVg using *pREF* control genes as negative scaling factors. To determine fold-change in *conu* coverage between conditions, read counts for each *conu* gene with multiple copies (cn>1) were compared to the *conu* gene with a single copy number (cn=1) between conditions. For example, the abundance of *conu1* (cn=1) from the control condition was compared to the abundance of *conu2* (cn=2), *conu3* (cn=3), *conu4* (cn=4) in the torkinib condition and vice versa. Differential expression of both *pREF* quantitative controls and human transcripts in tumor cells as a response to torkinib treatment was performed with edgeR (v3.30.3)[39].

## Mass spectrometry

*pREF* was transformed into chemically competent BL21 DE3 *E. coli* cells (NEB) according to manufacturer's protocols and was grown overnight at 37 °C on LB Kanamycin plates. Three separate colonies of each construct were spiked into 5 mL TB with 50 mg/mL Kanamycin and grown overnight at 37 °C, shaking at 200 rpm. Next, we expressed the constructs using 0.5 mM IPTG, and cultured them overnight at 30 °C, shaking at 200 rpm. After a 16 hour incubation, the *E. coli* cultures were pelleted at 20,000 g and protein was extracted using protein lysis buffer (5% SDS, 50 mM triethylammonium bicarbonate, 100 mM DTT,

pH 7.55), with cells disrupted using 4 cycles of boiling, ice and disruption using 0.1 mM glass beads in a Precellys 24 tissue homogeniser (Bertin Instruments).

*pREF* proteins may also be translated using an *E. coli* cell-free translation system by recipient laboratories. Cell-free translation efficiency can vary significantly between laboratories[27], and His-tag purified proteins should be checked on an SDS PAGE gel prior to use as a spike-in for Mass Spectrometry experiments.

Then 300 mg of extracted protein was digested with 1:50 Trypsin/Lys-C (Promega) on S-trap columns (Protifi), for 1 hour at 47ºC according to the manufacturer's instructions. Peptides were eluted by successively adding 80 μl of 5% acetonitrile in 0.1% formic acid, 80 μl of 50% acetonitrile in 0.1% aqueous formic acid and 80 μl of 75% acetonitrile in 0.1% formic acid with a 30 second centrifugation step at 4000 g between the addition of each elution buffer. The eluants were pooled, dried in a vacuum centrifuge and resuspended in 20 μl of buffer A (0.1% formic acid).

Samples were analysed using a Thermo Fisher Scientific Ultimate 3000 RSLC UHPLC and a Q-Exactive HF mass spectrometer. Samples were injected on a reverse-phase PepMap 100 C18 trap column (5 μm, 100 Å, 300 μm i.d. x 5 mm) at a flowrate of 15 μL/minute. After 3.0 minutes, the trap column was switched in-line with a Waters nanoEase M/Z Peptide CSH C18 resolving column (1.7 μm, 130 Å, 300 μm i.d. x 100 mm) and the peptides were eluted at a flowrate of 3 μL/minute using buffer A (0.1% formic acid) and buffer B (80 % acetonitrile in 0.1 % formic acid) as the mobile phases. The gradient consisted of: 8–2410% B for 0 to 6 minutes, 10–24% B from 6 to 43 minutes, 24–40% B from 43–51 minutes, 40-95% B from 51–57 minutes, followed by a wash, a return of 8% buffer B and equilibration prior to the next injection. The mass spectra were obtained in DIA mode with an MS1 resolution of 120,000, automatic gain control target at $3 \times 10^6$, maximum injection time at 200 ms and scan range from 400–1100 m/z. DIA spectra were recorded at resolution 30,000 and an automatic gain control target of $2 \times 10^5$. The 70 isolation windows were 10 m/z each from mass 405–1095.

### Mass Spectrometry analysis

Results were analysed in Spectronaut (15.2.210819.50606) using direct DIA analysis and default settings. Briefly, spectra were searched against the proco 1, 2 and 3 protein sequences and BL21 proteome with carbamidomethylation set as a fixed modification and methionine oxidation and N-terminal acetylation as variable with 1% false discovery rate cut-offs at the peptide spectral match, peptide and protein group levels. Quantitation was performed at the MS2 level with Q-value data filtering. Quantities of detected precursors were exported and summed to give total peptide quantities.

We differentiated the fully cleaved peptides from partially cleaved peptides, based on comparison to peptides predicted by Expasy[41] PeptideCutter set to Trypsin digestion. We calculated the proportion of fully cleaved peptides by summing the quantification of the fully cleaved peptides and comparing it to the quantification of all peptides (fully cleaved + partially cleaved) from each proco protein.

Next, we compared the detection of each fully cleaved peptide to their expected abundances. The majority of peptides were included as a single copy in each proco protein, and a subset of peptides were included at 2x, 4x and 8x. Peptide detection is expressed as a proportion of the total fully cleaved peptides for each protein, to normalise the detection of each protein between each of our three replicates.

We predicted the physicochemical properties of our peptides using the Expasy ProtParam tool[41], including Molecular Weight, theoretical pI, Instability Index, Aliphatic Index and Hydrophobicity. These were incorporated into a multiple linear regression in GraphPad Prism (v9.4.1), to determine the properties that best predicted relative quantification values. Peptide physicochemical properties were designated as fixed variables and relative quantification was the dependent variable.

As a comparison, we examined the peptides comprising six abundant *E. coli* proteins, comparing the observed and expected abundances. These were determined by ranking protein detection and selecting functionally diverse proteins that included >8 peptides that were long enough to be reliably detected by MS (>7 amino acids long). We conducted analyses of the relationship between peptide cleavage and observed vs expected abundance (in this case all peptides were included as a single copy), as described for our synthetic constructs (above).

### Reporting summary

Further information on research design is available in the Nature Portfolio Reporting Summary linked to this article.

## Data availability

The Next Generation Sequencing (NGS) data generated in this study have been deposited in the SRA database. The gDNA and translated RNA sequencing data were submitted with the PRJNA815898 BioProject Accession Identifier. Metagenomic data were submitted with the PRJNA781348 BioProject Accession Identifier, and the RNA-seq data were submitted with the PRJNA1073056 BioProject Accession Identifier. RNA-seq data were annotated based on the GENCODE primary assembly annotation v36 (https://www.gencodegenes.org/human/release_36.html). BAM alignments are available through DRYAD (10.5061/dryad.k0p2ngffn). The Mass Spectrometry DIA data generated in this study have been submitted to the PRIDE database, and are available via ProteomeXchange with identifier PXD035035 (doi:). Source data are provided with this paper.

## Code availability

All scripts and code used during the analysis are available from through Zenodo https://zenodo.org/records/10608215.

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

## Acknowledgements

We acknowledge the following funding sources: National Health and Medical Research Council (NHMRC grants APP1108254, APP1114016, APP1136067, awarded to T.R.M.), UNSW Tuition Fee Scholarship (TFS; to A.L.M.R), NHMRC Investigator Grant (2009010) to P.I.C., MRFF Investigator grant (MRF1173594) to I.W.D. and Cancer Institute NSW Early Career Fellowship 2018/ECF013 (to I.W.D). The contents of the published materials are solely the responsibility of the administering institution, a participating institution or individual authors, and they do not reflect the views of the NHMRC or CINSW.

## Author contributions

S.E.Y., P.I.C., E.M., N.S.S. and T.R.M. conceived the original experiment. S.Y, A.L.M.R and T.R.M. designed pREF. B.S.M., I.D., N.S.S. and I.S. performed NGS experiments. H.M.G. T.M. and E.M. performed Mass Spectrometry experiments. S.E.Y., H.M.G., T.W., A.C., S.K. and T.R.M. performed bioinformatic analysis. S.E.Y., H.M.G. and T.R.M. prepared the manuscript. All authors reviewed the manuscript.

## Competing interests

The Garvan Institute has filed patents covering aspects of this study. T.R.M. and H.M.G. have received financial support from Oxford Nanopore Technologies for travel, accommodations and research costs. The other authors declare no competing interests.
