## [Peer Review File · Nature Communications]

A universal molecular control for DNA, mRNA and protein expressionREVIEWER COMMENTS

Reviewer #1 (Remarks to the Author):

The manuscript of Youlten et al. is an interesting and well written description of a multigene synthetic plasmid intentionally designed to be used as a reference standard for DNA sequencing, RNA sequencing and proteomics. It contains high quality figures and a good explanation for the scope of the project and a fair discussion of the advantages and limitations of the work presented. It will be of interest to many of the journal's readers, especially due to the novel idea behind this work.

My own expertise in using reference standards materials for omics studies is limited so I can't fully comment on the quality of the presented standards in this paper, especially for the proteomics which is outside my comfort zone. My own expertise is on synthetic biology so I feel more inclined to comment on the design and workings of the plasmid made for this project, which largely I am happy with. My comments on the paper ahead of revisions are:

1. Calling a 10 kb plasmid a 'synthetic genome' is provocative and a bit of a stretch in my opinion. A genome is the information required for an organism or phage to propagate itself and I don't think this can really be called a genome by any definition. Can another name be considered? I should note that there is already a SynX chromosome published as part of the synthetic yeast genome project (<https://www.ncbi.nlm.nih.gov/pmc/articles/PMC5679077/>), so use of the name SynX is also confusing.

2. The most important aspects of an experimental standard is that it should behave predictably when used by different people in different labs and settings. This is almost entirely unaddressed in the work presented here. Key questions that jump out at me on this point relating to repeatability in different labs are:

(a) how stable is the plasmid over time in growth and culturing in E.coli, and purification for downstream use? Many synthetic plasmids mutate or recombine when cultured in E.coli. If this plasmid is shared with others to grow and use, how confident can we be that it works the same for everyone.

(b) how repeatable is the cell-free production of the RNAs and proteins in different groups? In my experience cell-free work has large batch variation effects and often is optimised by each group, rather than being done as a standard protocol.

I would appreciate in revisions that these aspects are addressed either experimentally or in the discussion of the results.

Reviewer #2 (Remarks to the Author):

The manuscript by Youlten et al. presents SynX, a new type of spike-in control that can be used as a reference across different sequencing (DNA, RNA) and proteomic analysis modalities. The study takes a synthetic biology approach to engineer the SynX genome, which is approx. 10,000 bases in length and includes various classes of control features that are present in different copy number, have different levels of repetitiveness or GC-content and can be in vitro transcribed or translated.

Specific comments and questions:

It was unclear to me which data was used for the various analyses, so maybe a table or figure panel on the experimental design and samples available can be added to the paper to make this easier to follow. For instance, in Figure 2, line 143 refers to 'DNA libraries' but it wasn't clear how many libraries (replicates) there were for Illumina and Nanopore and whether the figures show results from a representative library or results that are averages / summaries from all of the libraries? Similarly in Figure 3, it was unclear how the distributions shown in panel b arose? Is the data from multiple samples, and if so, how many? From what I understand there is 1 copy of conu 1x gene (yellow) so I was confused as to where the distribution comes from in b and the spread of

points for this control in c (right-hand panels). Similarly in Supp Figure S7a how does the pairing of points arise (same library split between sequencing technology or something else) and how many samples are there?

In the RNAseq data analysis, the authors find that conu genes allow better normalization with a method called RUVg relative to another popular tool TMM. Can this result be dug into a bit more i.e. what is the material effect on downstream analysis? Do you detect more genes as differentially expressed in this data set after the conu gene normalization? What about if you use other endogenous (human) genes as controls in place of the conu genes in the RUVg analysis - are results better or worse? The assessment of improvement was very light on in this section in my opinion.

Overall, I envisage that these versatile controls could be very useful to the genomics research community. I wondered how they would be shared and distributed to others who wanted to make use of them? This was unclear from the manuscript (beyond making the sequence publicly available).

Line 40 Check expression 'technical that confounds'

Line 41, 42, 45 'phiX-174' -> 'PhiX-174' (for consistency)

Line 51 Check expression 'spiked-in' into -> 'spike-in' to?

Line 63 Check expression 'and the integrate'

Line 71 Typo 'stnadards'

Line 73 'matched mRNA and protein controls independently,' (missing word / phrase at end of this clause - maybe prepared?)

Line 165 '~16% than expected' missing word (shorter?)

Many software / analysis references in the Methods appear as PMIDs and don't make their way into the reference list - please correct this

Line 330 Delete extra ,

Line 437, 446, 506 'libraries were...' - how many were there for each data set?

Line 453 'RNA libraries' - which ones? (the ones the section that begins on line 494? If so, maybe bring this section forward) and how were they prepared and sequenced (i.e. what kits / chemistry)?

Line 639 Check author name formatting for ref 18.

Reviewer #3 (Remarks to the Author):

In this manuscript, the authors describe a synthetic genome "SynX", which can be transcribed and translated in vitro to simultaneously produce DNA, RNA, and protein reference standards within one integrated system. SynX contains various sequences with different properties. Conu genes are present in different replicate numbers, which should be reflected in DNA or RNA sequencing results. Repe genes show a high content of repeat sequences, while gece genes contain a high GC content. The correct representation of both types of sequences is challenging for current sequencing methods. Proco genes can be translated and transcribed into proteins containing different copies of tryptic peptides. These peptides then could serve as quantitative reference in LC-MS/MS proteomic measurements. Additionally, the SynX genome contains recognition sites for commonly used restriction enzymes and can be cleaved to produce a DNA fragment ladder with defined fragment sizes.

According to the results, the SynX genome can be used well as a qualitative and quantitative reference for DNA sequencing, as is nicely shown in a comparison of Illumina and ONT sequencing technologies. Cell-free, in vitro transcription of SynX resulted in a lower quantitative accuracy of conu gene expression, which makes this technology less useful for standardization on RNA level. The offered explanation that hairpin formation within the RNA might be a confounding factor is plausible, but this observation limits the applicability of SynX more applicable as a standard for RNA sequencing. Finally, the expression of proco genes in E. coli and subsequent LC-MS/MS measurements of the resulting proteins showed a very poor correlation of theoretical peptide copy numbers to actually measured peptides, hence further limiting the applicability of SynX as a

proteomics standard. On the other hand, unexpected (or possibly unknown at present) regulatory events in gene expression are a common feature in cell biology and synthetic genomes offer an approach to better understanding this topic. In this regard, a more detailed approach to explaining the aforementioned discrepancies would be of interest.

- The authors claim (Abstract) that the SynX system can be sustainably prepared by recipient laboratories using common molecular biology techniques. So far it apparently has only been tested in *E. coli* and its performance in other cells remains unclear. It is also unclear, whether other laboratories have received and tested SynX. The authors also do not show how to practically use SynX in a cell-culture experiment. I

- The authors have included all possible 6-mer nucleotide combinations (4096 in total) into the SynX genome. In line 88 the authors confirm that they have tested the SynX system to have no sequence homology greater than 18 nucleotides to natural genome sequences. It remains unclear which natural genomes were used as comparisons and how the claim that SynX can thus be used as a reference standard for any organism is supported (line 90).

- Since all possible combinations of 6-mer nucleotide sequences are included into the genome, many recognition sites of restriction enzymes like *EcoRI* should be included into the main genome as well. Was this taken into account when creating DNA fragment ladders (line 91)?

- In some cases, sentences lack defining words. E.g. the sentence in line 165 doesn't indicate whether these 16% are more, less, or equal to what was expected.

- The color coding in figure 3d (green / blue) is not clear.

- In line 308, the authors suggest that poor correlation of proco and *E. coli* peptide abundance to the measured signal might stem from low accuracy of LC-MS/MS DIA quantitation. Mass-spectrometry intrinsic effects, like ionization efficiency and ion suppression, play a major role here. Peptides of different sequences cannot be compared quantitatively, because they might show entirely different physicochemical behavior i.e. ionization efficiency. For this reason, different peptides of *E. coli* housekeeping proteins also show different abundances, when compared to each other (Fig. S11). Across samples, only peptides of the same sequence can be used for quantitative comparison (preferably on MS1 level). In this context it would be interesting to know, whether and which protein libraries were used for the DIA analysis.

Other reasons might be e.g. faulty translation, post-translational modifications, degradation, or cellular export of proteins within the *E. coli* expression system.

Reviewer #1 (Remarks to the Author):

The manuscript of Youlten et al. is an interesting and well written description of a multigene synthetic plasmid intentionally designed to be used as a reference standard for DNA sequencing, RNA sequencing and proteomics. It contains high quality figures and a good explanation for the scope of the project and a fair discussion of the advantages and limitations of the work presented. It will be of interest to many of the journal's readers, especially due to the novel idea behind this work.

My own expertise in using reference standards materials for omics studies is limited so I can't fully comment on the quality of the presented standards in this paper, especially for the proteomics which is outside my comfort zone. My own expertise is on synthetic biology so I feel more inclined to comment on the design and workings of the plasmid made for this project, which largely I am happy with. My comments on the paper ahead of revisions are:

1. Calling a 10 kb plasmid a 'synthetic genome' is provocative and a bit of a stretch in my opinion. A genome is the information required for an organism or phage to propagate itself and I don't think this can really be called a genome by any definition. Can another name be considered? I should note that there is already a SynX chromosome published as part of the synthetic yeast genome so use of the name SynX is also confusing.

Thank you for your comment. The design of our SynX was initially inspired from the *PhiX-174* genome (which is ~5kb in size), and we accordingly tried to adopt similar nomenclature. However, we agree that this may be confusing, particularly given the *SynX* chromosome used in the Yeast 2.0 project.

Therefore, we propose to adopt similar nomenclature as for plasmids, using this helpful blog:

https://blog.addgene.org/plasmids-101-how-to-name-your-plasmid-in-3-easy-steps?gclid=CjwKCAiAkfucBhBBEiwAFjbr8LyOxtknipD18EosxbQ6qyow83-NfF6NJK59RSWh1FRxHJQugaTsBoCl_gQAvD_BwE

Following this convention, we propose to change the name to ***pREF***, for reference control applications and no longer refer to *pREF* as a genome, but rather a plasmid.

2. The most important aspects of an experimental standard is that it should behave predictably when used by different people in different labs and settings. This is almost entirely unaddressed in the work presented here. Key questions that jump out at me on this point relating to repeatability in different labs are:

(a) how stable is the plasmid over time in growth and culturing in *E. coli*, and purification for downstream use? Many synthetic plasmids mutate or recombine when cultured in *E. coli*. If this plasmid is shared with others to grow and use, how confident can we be that it works the same for everyone.

We agree that an experimental standard should behave predictably when used at different times or sites. Whilst the *PhiX-174* genome has demonstrated how a ~5kb genome can be propagated and used in next-generation sequencing, we agree that our reference plasmid has not yet been tested under so many different labs and settings.

Therefore, we have now highlighted in the text that the *pREF* will need to be transformed and propagated into a stable *E. coli* line and requires DNA sequencing confirmation prior to *in vitro* transcription and translation:

“These regulatory elements enable ongoing and sustainable production of *pREF* by independent laboratories through transformation and propagation in a stable *E. coli* line and purification using standard plasmid preparation techniques (see **Methods**).”

Additionally, the following has been added to the Methods section:

“For laboratories propagating *pREF*, we recommend DNA sequencing prior to translation and its use as a sequencing standard. Although spontaneous deletion errors are observed less frequently in stable *E. coli* lines, they can occur in repetitive sequences, such as those included in *pREF*.”

(b) how repeatable is the cell-free production of the RNAs and proteins in different groups? In my experience cell-free work has large batch variation effects and often is optimised by each group, rather than being done as a standard protocol. I would appreciate in revisions that these aspects are addressed either experimentally or in the discussion of the results.

We have not yet performed the *in vitro* translation in different laboratories and don't yet understand the variation between laboratories. However, we would like to highlight to the reviewer that *pREF* plasmid now provides a standardised reference template to perform these comparisons between laboratories. To address these aspects in the discussion of the results, we have included the following Results section:

“*pREF* proteins may also be translated using an *E. coli* cell-free translation system by recipient laboratories. Cell-free translation efficiency can vary significantly between laboratories²¹, and His-tag purified proteins should be checked on an SDS PAGE gel prior to use as a spike-in for Mass Spectrometry experiments.”

And in the Discussion section:

“As cell-free translation rates can vary due to systematic and stochastic variables (Chizzolini et. al., 2017), recipient laboratories should verify proco protein quantity prior to their use as spike-ins.”

Reviewer #2 (Remarks to the Author):

The manuscript by Youlten et al. presents SynX, a new type of spike-in control that can be used as a reference across different sequencing (DNA, RNA) and proteomic analysis modalities. The study takes a synthetic biology approach to engineer the SynX genome, which is approx. 10,000 bases in length and includes various classes of control features that are present in different copy number, have different levels of repetitiveness or GC-content and can be *in vitro* transcribed or translated.

Specific comments and questions:

It was unclear to me which data was used for the various analyses, so maybe a table or figure panel on the experimental design and samples available can be added to the paper to make this easier to follow. For instance, in Figure 2, line 143 refers to 'DNA libraries' but it wasn't clear how many libraries (replicates) there were for Illumina and Nanopore and whether the figures show results from a representative library or results that are averages / summaries from all of the libraries?

To provide an overview of the experimental samples, we have included a new Supplementary Figure that gives an overview of the samples used in each sequencing experiments (**Fig. S14**). This new Figure is now cited in the text as follows (e.g.):

“To demonstrate this approach, we first sequenced four technical replicates of *pREF* using Illumina short-read and Oxford Nanopore (ONT) long-read sequencing (**Fig. S14**).”

Similarly in Figure 3, it was unclear how the distributions shown in panel b arose? Is the data from multiple samples, and if so, how many? From what I understand there is 1 copy of *conu* 1x gene (yellow) so I was confused as to where the distribution comes from in b and the spread of points for this control in c (right-hand panels).

Figure 3 was prepared using a single replicate. The separate data points are due to our use of a 31-nucleotide sliding window approach in our analyses. Copy numbers were calculated for each 31-nucleotide sliding window of each of the four *conu* genes and compared to the expected copy numbers.

There were single replicates included in each analysis. This has now been clarified in the text;

“Density histogram from k-mer counts for *conu* gene families illustrates the distribution of technical variation in 31-mer normalised read count, calculated using a sliding window approach.”

And we have included the following discussion;

“To demonstrate this approach, we analysed the read counts across 31-mer sliding windows for all *conu* genes in Illumina and ONT sequenced DNA libraries (Fig. S14, Fig. 3b,c, Fig. S7a).”

Similarly in Supp Figure S7a how does the pairing of points arise (same library split between sequencing technology or something else) and how many samples are there?

Figure S7a shows ONT and Illumina sequencing data generated from single replicates. The separate data points are due to our use of a 31-nucleotide sliding window approach in our analyses. Normalised numbers were calculated for each 31-nucleotide sliding window of each of the four *conu* genes and compared to the expected copy numbers.

This has now been clarified in the text;

“Scatter-plot compares the measurement of normalised gene counts across 31-mer sliding windows for *conu1-4* genes, in Illumina and ONT sequencing libraries.”

In the RNAseq data analysis, the authors find that *conu* genes allow better normalization with a method called RUVg relative to another popular tool TMM. Can this result be dug into a bit more i.e. what is the material effect on downstream analysis? Do you detect more genes as differentially expressed in this data set after the *conu* gene normalization? What about if you use other endogenous (human) genes as controls in place of the *conu* genes in the RUVg analysis - are results better or worse? The assessment of improvement was very light on in this section in my opinion.

The use of normalisation techniques, including spike-in controls, has been extensively analysed in the literature (including Risso et al., 2014, Reis et al., 2020). However, we did want to highlight that our *pREF* controls could be similarly used for normalisation, with the following sentence added to the Results section;

“RUVg can remove technical variation through factor analysis using reference standards¹⁶. We tested whether RUVg normalisation with *pREF* could also be used for normalisation of an RNA-seq experiment.”

And we also included the following discussion in;

“This spike-in use of *conu* mRNA control genes allows normalisation between RNA sequencing libraries, without relying on housekeeping genes, which can be highly variable (Ruan et. al., 2007). This method will be particularly useful for experiments that induce global gene expression changes or with low biological replication, where negative controls and analyses of variation are not an option.”

Overall, I envisage that these versatile controls could be very useful to the genomics research community. I wondered how they would be shared and distributed to others who wanted to make use of them? This was unclear from the manuscript (beyond making the sequence publicly available).

The *pREF* plasmid will be submitted and distributed by Addgene (<https://www.addgene.org/>) and the following text has now been added to the Methods and Results sections:

“*pREF* has been deposited within the Addgene catalog for access and distribution.”

Line 40 Check expression ‘technical that confounds’
Line 41, 42, 45 ‘phiX-174’ -> ‘PhiX-174’ (for consistency)
Line 51 Check expression ‘spiked-in’ into -> ‘spike-in’ to?
Line 63 Check expression ‘and the integrate’
Line 71 Typo ‘stnadards’

Line 73 'matched mRNA and protein controls independently,' (missing word / phrase at end of this clause – maybe prepared?)
Line 165 '~16% than expected' missing word (shorter?)
Line 330 Delete extra ,
Line 437, 446, 506 'libraries were...' – how many were there for each data set?
Line 453 'RNA libraries' – which ones? (the ones the section that begins on line 494? If so, maybe bring this section forward) and how were they prepared and sequenced (i.e. what kits / chemistry)?
Line 639 Check author name formatting for ref 18.

We thank the reviewer for identifying these errors that have now been corrected in the manuscript.

Many software / analysis references in the Methods appear as PMIDs and don't make their way into the reference list – please correct this

Thanks for highlighting this oversight. These software/analysis methods have now been added to the reference section.

Reviewer #3 (Remarks to the Author):

In this manuscript, the authors describe a synthetic genome "SynX", which can be transcribed and translated in vitro to simultaneously produce DNA, RNA, and protein reference standards within one integrated system. SynX contains various sequences with different properties. *Conu* genes are present in different replicate numbers, which should be reflected in DNA or RNA sequencing results. *Repe* genes show a high content of repeat sequences, while *gece* genes contain a high GC content. The correct representation of both types of sequences is challenging for current sequencing methods. *Proco* genes can be translated and transcribed into proteins containing different copies of tryptic peptides. These peptides then could serve as quantitative reference in LC-MS/MS proteomic measurements. Additionally, the SynX genome contains recognition sites for commonly used restriction enzymes and can be cleaved to produce a DNA fragment ladder with defined fragment sizes.

According to the results, the SynX genome can be used well as a qualitative and quantitative reference for DNA sequencing, as is nicely shown in a comparison of Illumina and ONT sequencing technologies. Cell-free, in vitro transcription of SynX resulted in a lower quantitative accuracy of *conu* gene expression, which makes this technology less useful for standardization on RNA level. The offered explanation that hairpin formation within the RNA might be a confounding factor is plausible, but this observation limits the applicability of SynX more applicable as a standard for RNA sequencing.

We disagree that SynX is more limited during RNA sequencing, particularly in comparison to standard methods using normalisation with housekeeping genes. In our study, we performed the synthesis of the *conu* genes, with the resulting mRNA being used in both the ONT and Illumina sequencing experiments. Therefore, these *conu* mRNAs enable a direct comparison between the sequencing technologies and the lower quantitative accuracy of gene expression, highlighted by the review, is likely due to greater technical variation during RNA sequencing, which includes cDNA Reverse Transcription and PCR amplification steps.

The advantage of SynX is that it enables users to detect and measure this technical variation of RNA sequencing which is greater than DNA sequencing. We highlight RNA hairpins as an additional contributing source of variation during RNA sequencing following our analysis of *conu* genes. To clarify this, we have included the following text to the manuscript:

"This spike-in use of conu mRNA controls allows normalisation between RNA sequencing libraries, without relying on housekeeping genes, which can be highly variable (Ruan et. al., 2007). This method will be particularly useful for experiments that induce global gene expression changes or with low biological replication, where negative controls and analyses of variation are not an option."

Finally, the expression of *proco* genes in *E. coli* and subsequent LC-MS/MS measurements of the resulting proteins showed a very poor correlation of theoretical peptide copy numbers to actually measured peptides, hence further limiting the applicability of SynX as a proteomics standard. On the other hand, unexpected (or

possibly unknown at present) regulatory events in gene expression are a common feature in cell biology and synthetic genomes offer an approach to better understanding this topic. In this regard, a more detailed approach to explaining the aforementioned discrepancies would be of interest.

We disagree that poor quantitative correlation of LC-MS/MS measurements of *proco* genes represents a limitation of the *SynX* as a proteomic standard. Instead, we propose this example illustrates how the *SynX* proteomic standard can detect the poor quantitative performance of LC-MS/MS. The detection of peptides using MS may be impacted by many technical and biological variables, resulting in low quantitative accuracy. Our study explores these variables that may impact peptide detection (see **Supplementary Figure 12**). In addition, we have included a statement about the further biological variables that may impact peptide detection;

“Further variables such as post-translational modifications, degradation or cellular export of proteins may also have impacted peptide detection, however their measurement is outside the scope of this experiment.”

- The authors claim (Abstract) that the *SynX* system can be sustainably prepared by recipient laboratories using common molecular biology techniques. So far it apparently has only been tested in *E. coli* and its performance in other cells remains unclear. It is also unclear, whether other laboratories have received and tested *SynX*. The authors also do not show how to practically use *SynX* in a cell-culture experiment.

We thank the reviewer for their comment. The independent propagation of the *SynX* is an advantage for easy distribution compared to previous reference standards. However, the plasmid backbone (pUC) used for the *SynX* is designed for preparation in *E. coli* competent cells, and is not intended to be propagated in other cells. The protocol for propagation uses standard inoculation, amplification and extraction steps, and is described in detail in materials and methods, and the following text has been added to the Results:

“These regulatory elements enable ongoing and sustainable production of *pREF* by independent laboratories through transformation and propagation in a stable *E. coli* line and purification using standard plasmid preparation techniques (see **Methods**).”

Additionally, the following has been added to the Methods:

“The plasmid backbone used for *pREF* is only suitable for propagation in stable *E. coli*, and not mammalian cell lines.”

- The authors have included all possible 6-mer nucleotide combinations (4096 in total) into the *SynX* genome. In line 88 the authors confirm that they have tested the *SynX* system to have no sequence homology greater than 18 nucleotides to natural genome sequences. It remains unclear which natural genomes were used as comparisons and how the claim that *SynX* can thus be used as a reference standard for any organism is supported (line 90).

To analyse the homology of the *SynX*, we compared the sequence to the NCBI Nucleotide Collection (nr/nt) which contains all RefSeq RNA records plus all GenBank sequences except for those from the EST, GSS, STS and HTG divisions. We excluded any sequences that had homology greater than 18nt with any sequence in this NCBI Nucleotide Collection (nr/nt). As a result, any sequenced alignment greater than 18nt generated from the *SynX* sequence can be distinguished from any natural genetic sequence. Therefore, we claim that the *SynX* can be used as a reference standard for any organism that is included in the NCBI Nucleotide Collection (nr/nt). This has now been clarified in Methods and Materials as follows:

“Furthermore, the synthetic genes have no homology (greater than 18nt) to natural gene sequences included in the NCBI nr/nt database and can be easily distinguished from natural RNA or DNA samples, allowing the use of *pREF* as a spike-in control in the study of any organism included in the NCBI nr/nt database¹⁰.”

- Since all possible combinations of 6-mer nucleotide sequences are included into the genome, many recognition sites of restriction enzymes like *EcoRI* should be included into the main genome as well. Was this taken into account when creating DNA fragment ladders (line 91)?

Yes, following design, a number of 6-mers were excluded to avoid homology to Restriction Enzymes. This has now been clarified in the Results section as follows:

“*pREF* includes a full representation of all possible 6-mers (excluding unintended Restriction Enzyme recognition sequences), thereby providing a comprehensive evaluation of sequencing accuracy under different nucleotide contexts (Fig. S1a).”

- In some cases, sentences lack defining words. E.g. the sentence in line 165 doesn't indicate whether these 16% are more, less, or equal to what was expected.

Thanks for highlighting these cases that have now been clarified in the text as follows:

“The majority (93%) of errors at repeats were deletions, resulting in reads being ~16% shorter than expected (Fig. S6c).”

- The color coding in figure 3d (green / blue) is not clear.

We have added extra labels to **Figure 3d** to clarify the colour coding.

- In line 308, the authors suggest that poor correlation of *proco* and *E. coli* peptide abundance to the measured signal might stem from low accuracy of LC-MS/MS DIA quantitation. Mass-spectrometry intrinsic effects, like ionization efficiency and ion suppression, play a major role here. Peptides of different sequences cannot be compared quantitatively, because they might show entirely different physicochemical behavior i.e. ionization efficiency. For this reason, different peptides of *E. coli* housekeeping proteins also show different abundances, when compared to each other (Fig. S11). Across samples, only peptides of the same sequence can be used for quantitative comparison (preferably on MS1 level). In this context it would be interesting to know, whether and which protein libraries were used for the DIA analysis. Other reasons might be e.g. faulty translation, post-translational modifications, degradation, or cellular export of proteins within the *E. coli* expression system.

We agree that mass-spectrometry intrinsic effects, such as ionization efficiency and ion suppression, play a major role in confounding quantitative measurements, and only peptides of the same sequence can be used for quantitative comparison across samples. Indeed, we observe a low between-replicate variability for the quantification of individual peptides between samples, as expected, alongside high variation between peptides in the same sample. This limitation of LC-MS/MS limits the analysis and insight that the *proco* genes can provide in our proteomics study.

Nevertheless, we see the impact of these variables on the expression of the *proco* proteins and provide a detailed analysis of the potential role of peptide physicochemical properties in **Supplementary Figure 12**. We compare this analysis to protein samples extracted from BL21 DE3 *E. coli* that were used for the DIA analyses. However, we are unable to profile the post-translational modifications, degradation, or export of the *proco* proteins within the scope of this study.

This limitation is included in the Results as follows:

“Further variables such as post-translational modifications, degradation or cellular export of proteins may also have impacted peptide detection, however their measurement is outside the scope of this experiment.”

REVIEWER COMMENTS

Reviewer #1 (Remarks to the Author):

Youlten, Gunter et al, have revised their manuscript into an improved version that I feel is now ready for publication. Looking through the changes made in response to my original points and those made by the other reviewers, makes me more confident that the claims in this work are supported by the results and the description of the research. I thank the authors for improving the clarity of their manuscript, particularly in terms of its motivations and limitations.

It is still a bit disappointing that the work doesn't show any evidence of the pREF being used as a standard that works between different labs/institutions, which was one of my key recommendations. However, I can appreciate that asking another lab to repeat your work as part of a paper is not straightforward and will add significant extra time and cost to the authors to add this to their study.

Reviewer #2 (Remarks to the Author):

The have authors have adequately addressed each of my questions in their revised manuscript.

Reviewer #3 (Remarks to the Author):

The authors have addressed most concerns. However, they falsely interpret the LC-MS/MS data.

From a plasmid SynX, artificial proteins are translated that contain multiple, repeating sequences for tryptic peptides. The idea is, that after digestion there are 1, 2, 4, or 8 times the amount of these peptides in the sample. However, the peptides have different sequences and one can see very well in the data that some of them ionize well, others not at all and that a direct quantitative comparison of peptides with different sequences is not possible. Apart from that, the peptides are not optimized in the first place to be detectable in MS at all. Unsurprisingly, we see that the detected MS signals do not match the expected peptide copy numbers at all. Of course, quantification across three technical replicates was still stable across three technical replicates. The authors' conclusion is that MS-based proteomics inherently and systematically fails to quantify what they would have demonstrated with their plasmid system. Yet, their data only corroborates the know fact that MS-based intensity of peptides also depends on their sequence. Hence, this reviewer cannot support their conclusion of " poor quantitative performance of LC-MS/MS". This reviewer does not support publication of this manuscript in the present form. The LC-MS/MS data must be addressed better.

REVIEWER COMMENTS

We would like to express our gratitude to the reviewers for their valuable time and effort devoted to evaluating our manuscript which has substantially enhanced the quality of our work.

Reviewer #1 (Remarks to the Author):

Youlten, Gunter et al, have revised their manuscript into an improved version that I feel is now ready for publication. Looking through the changes made in response to my original points and those made by the other reviewers makes me more confident that the claims in this work are supported by the results and the description of the research. I thank the authors for improving the clarity of their manuscript, particularly in terms of its motivations and limitations.

It is still a bit disappointing that the work doesn't show any evidence of the *pREF* being used as a standard that works between different labs/institutions, which was one of my key recommendations. However, I can appreciate that asking another lab to repeat your work as part of a paper is not straightforward and will add significant extra time and cost to the authors to add this to their study.

We are grateful to the reviewer for their comments and concur that employing *pREF* for accurate inter-laboratory comparisons is indeed a significant benefit of this approach. We have initiated collaboration with multiple laboratories to evaluate the SynX control for assessing inter-laboratory variation and to foster further adoption. However, we acknowledge that these efforts are beyond the scope of the present manuscript.

Reviewer #2 (Remarks to the Author):

The have authors have adequately addressed each of my questions in their revised manuscript.

We thank the reviewer for their time and efforts.

Reviewer #3 (Remarks to the Author):

The authors have addressed most concerns. However, they falsely interpret the LC-MS/MS data.

From a plasmid SynX, artificial proteins are translated that contain multiple, repeating sequences for tryptic peptides. The idea is, that after digestion there are 1, 2, 4, or 8 times the amount of these peptides in the sample. However, the peptides have different sequences and one can see very well in the data that some of them ionize well, others not at all and that a direct quantitative comparison of peptides with different sequences is not possible. Apart from that, the peptides are not optimized in the first place to be detectable in MS at all. Unsurprisingly, we see that the detected MS signals do not match the expected peptide copy numbers at all. Of course, quantification across three technical replicates was still stable across three technical replicates. The authors' conclusion is that MS-based proteomics inherently and systematically fails to quantify what they would have demonstrated with their plasmid system. Yet, their data only corroborates the know fact that MS-based intensity of peptides also depends on their sequence. Hence, this reviewer cannot support their conclusion of "poor quantitative performance of LC-MS/MS". This reviewer does not support publication of this manuscript in the present form. The LC-MS/MS data must be addressed better.

We appreciate the reviewer's evaluation of our manuscript and their concerns regarding the quantitative performance of LC-MS/MS based on measurement using our *pREF* plasmid system. We understand that peptides with varying sequences and ionization efficiencies confound quantitative analysis, and that a direct quantitative comparison of peptides with different sequences is not possible. In light of these concerns, we have revised our manuscript to provide a more balanced interpretation of our findings and clarify that the observed discrepancies in detected MS signals may

be attributed to the inherent sequence-dependent variation in ionization efficiency, rather than a systematic failure of LC-MS/MS quantification.

To address these concerns, we have included the following revision of the results:

“We first used the *proco* genes to evaluate the quantitative accuracy of LC-MS/MS DIA datasets by comparing the measured abundance of individual *proco* peptides relative to their expected copy numbers (Fig. 5c). We found that observed *proco* peptide abundance correlated poorly with expected copy numbers ($R^2=3.057 \times 10^{-6}$; Fig. 5d). These observed discrepancies in detected MS signals may be attributed to the sequence-dependent variation in ionization efficiencies between different peptides. However, despite these limitations, we observed high reproducibility in peptide quantification across three technical replicates indicating that while LC-MS/MS may not provide accurate quantification between peptides with different sequences, it does offer a reproducible and quantitative means of assessing the relative abundance of a given peptide between different samples (Fig. 5b).”

As well as:

“We also considered whether the observed discrepancies in LC-MS/MS quantification measured with the *proco* peptide controls similarly impacted the measurements of the accompanying *E. coli* proteins. To evaluate the variation in quantification, we assumed that each unique peptide within the selected *E. coli* housekeeping proteins was present once and was therefore equally abundant (Fig. S11a). Therefore, by comparing the variation in peptide quantification within *E. coli* proteins, we could estimate the quantitative accuracy of our LC-MS/MS experiment and compare it to *proco* controls. Similar to our *proco* peptide results, we observed a wide variation in the quantification of different *E. coli* peptides, with strong reproducibility for the same peptide between biological replicates (Fig. S11b-c). This demonstrates how *proco* proteins can evaluate and validate the quantification of protein abundance using MS methods.”

And the following additions to the Discussion:

“However, our analysis found LC-MS/MS exhibited discrepancies in quantification between individual *pREF* proteins, and supports the well-established finding that peptides do not ionize equally in LC-MS/MS. This renders direct quantitative comparison of peptides with differing sequences infeasible and will likely require further advances, such as the implementation of machine-learning models that predict and normalize intensity response based on sequence, to improve quantitative comparisons between peptides using LC-MS/MS^{20,21}. These sequence-dependent discrepancies in peptide quantification likely contribute to the poor correlation often reported between the transcriptome and proteome¹, and demonstrate how *pREF* can help integrate DNA, RNA sequencing and mass spectrometric experiments that are becoming increasingly used in multi-omic studies^{22,23}.”

Despite these limitations, we do observe that the quantification of individual *pREF* peptides is highly reproducible, providing a reliable means of assessing the relative abundance of a given protein between different samples.”

REVIEWERS' COMMENTS

Reviewer #3 (Remarks to the Author):

The revised manuscript is now factually correct and addresses LC-MS/MS based proteomics' limitation that only identical peptide sequences can be directly compared quantitatively. Although this precludes pREFs ability to function as an integrated gene expression standard across genome, transcriptome and proteome, expressed proteins might very well be used to compare technical replicates of different MS runs. For this, however, other standards have been already developed and are commonly used.